# FunnyNodules: A Customizable Medical Dataset Tailored for Evaluating Explainable AI

**Luisa Gallée** [1,2] 🆔                                          LUISA.GALLEE@UNI-ULM.DE
[1] *Experimental Radiology, Ulm University Medical Center, Ulm, Germany*
[2] *XAIRAD - Cooperation for Artificial Intelligence in Experimental Radiology, Ulm, Germany*

**Yiheng Xiong** [1,2] 🆔

**Meinrad Beer** [2,3] 🆔
[3] *Department of Diagnostic and Interventional Radiology, Ulm University Medical Center, Ulm, Germany*

**Michael Götz** [1,2] 🆔

**Editors:** Accepted for publication at MIDL 2026

## Abstract

Densely annotated medical image datasets that capture not only diagnostic labels but also the underlying reasoning behind these diagnoses are scarce. Such reasoning-related annotations are essential for developing and evaluating explainable AI (xAI) models that reason similarly to radiologists: making correct predictions for the right reasons. To address this gap, we introduce *FunnyNodules*, a fully parameterized synthetic dataset designed for systematic analysis of attribute-based reasoning in medical AI models. The dataset generates abstract lung nodule–like shapes with controllable visual attributes such as roundness, margin sharpness, and spiculation. The target class is derived from a predefined attribute combination, allowing full control over the decision rule that links attributes to the diagnostic class. We demonstrate how FunnyNodules can be used in model-agnostic evaluations to assess whether models learn correct attribute–target relations, to interpret over- or underperformance in attribute prediction, and to analyze attention alignment with attribute-specific regions of interest. The framework is fully customizable, supporting variations in dataset complexity, target definitions, class balance, and beyond. With complete ground truth information, FunnyNodules provides a versatile foundation for developing, benchmarking, and conducting in-depth analyses of explainable AI methods in medical image analysis.

**Keywords:** Dataset, Explainable AI, Evaluation

## 1. Introduction

In medical image analysis, numerous machine learning models and explainable AI (xAI) methods have been proposed (Wang et al., 2022; Shi et al., 2023; Frasca et al., 2024; Bhati et al., 2024; Muhammad and Bendechache, 2024). While model performance is typically well evaluated, other aspects, such as the correctness of model reasoning, that is, whether a model makes the right decision for the right reasons, are often insufficiently assessed (Rudin, 2019). Even for xAI methods that are explicitly designed to improve interpretability, systematic evaluation remains a major challenge (Muhammad and Bendechache, 2024). A key

reason for this limitation is the lack of comprehensive ground truth for attributes and explanations, which is necessary to systematically evaluate model reasoning and interpretability. In addition to annotations for the target classes, the evaluation of xAI methods also requires ground truth annotations for visual explanations at the sample level. Such annotations are rare, particularly in the medical domain where dataset sizes are inherently limited (Litjens et al., 2017).

To address this gap, we introduce FunnyNodules, a synthetic image dataset specifically designed for evaluating AI and xAI methods in medical imaging. A core feature of FunnyNodules is its comprehensive annotation. Since the image generation process is explicitly controlled, all samples follow predefined appearance features (referred to as attributes), similar to the synthetic datasets FunnyBirds (Hesse et al., 2023) and Elements (Nicolson et al., 2025). While FunnyBirds applies this concept to natural images using discrete object parts (e.g., bird beak or wing) as features, Elements generates abstract graphical primitives such as squares and circles with varying colors and patterns. In contrast, FunnyNodules is tailored to the medical imaging domain, modeling radiology-specific attributes such as intensity, shape, and margin characteristics. The parameterized generative approach enables the automatic creation of complete ground truth information, including target class and attribute labels as well as region-of-interest (ROI) masks, without being affected by inter- or intra-rater variability.

Furthermore, the FunnyNodules framework provides a high degree of customization. Both the variability in image appearance and the decision rules for target classification can be adapted to different levels of complexity. This enables researchers to tailor the dataset to the specific evaluation needs of their models. In contrast to existing synthetic datasets generated using large generative models such as diffusion models (Pinaya et al., 2022; Khosravi et al., 2024; Gallée et al., 2026) or GAN (Frid-Adar et al., 2018), the goal of FunnyNodules is not to simulate realistic data. Instead, it focuses on simulating attribute relationships with clearly defined ground truth and full controllability. In addition, this approach does not require any real training data and is free from data-driven biases.

FunnyNodules provides a controlled environment for evaluating AI models and xAI methods in medical image analysis, allowing the investigation of model reasoning behavior without the limitations of real-world datasets. In this paper, we

- introduce FunnyNodules, a synthetic vision dataset inspired by medical image interpretation and designed for systematic evaluation of AI models as explainability methods.

- demonstrate how FunnyNodules can be used to assess model behavior with respect to attribute sensitivity, reasoning correctness, and trustworthiness.

The dataset generation code and all presented experiments are publicly available at https://github.com/XRad-Ulm/FunnyNodules.

## 2. Dataset

The aim of this synthetic dataset is to enable a comprehensive evaluation of AI models for medical images. Full control over the image generation process allows the creation

of depictions that follow defined visual attributes. This approach offers a high degree of customizability, scalability, and targeted adjustments, providing strong flexibility for model-agnostic evaluations.

The FunnyNodules framework is based on the idea of representing abstract lung nodules through controllable visual attributes. Rating the malignancy of lung nodules is a clinically relevant and well-studied task in radiology (Furuya et al., 1999; El-Baz et al., 2013). The diagnostic evaluation strongly depends on quantifiable visual features, such as intensity, roundness, and margin sharpness (Armato III et al., 2015, 2011), which directly correspond to the concept of attributes in explainable AI.

## 2.1. FunnyNodules

The FunnyNodules dataset, employed in the experiments of this study, comprises abstract nodules described by six visual attributes:

- roundness            (r)     1-*round*, 5-*oval*
- spiculation          (sp)    1-*none*, 5-*marked*
- edge sharpness       (es)    1-*sharp*, 5-*soft*
- size                 (s)     1-*small*, 5-*big*
- intensity            (i)     1-*dark*, 5-*bright*
- internal structure   (is)    0-*absent*, 1-*present*

The target class is defined based on combinations of these attributes (see Algorithm 1) and is ordinal, ranging from 1 to 5, the same scale used for all attributes except internal structure, which is binary. However, a major advantage of the synthetic FunnyNodules framework is its ability to implement different scales and rules as desired.

For performance evaluation, we use the Within-1-Accuracy metric, following prior attribute-based studies (LaLonde et al., 2020; Gallée et al., 2023). Predictions are considered correct if they deviate by at most $\pm 1$ from the ground truth score for ordinal labels, whereas internal structure requires exact agreement.

To provide insight into the distribution of attribute and target values in FunnyNodules, a histogram analysis is provided in Appendix A.1.

### 2.1.1. IMAGE GENERATION

Each image in the FunnyNodules dataset is synthetically generated through a parameterized algorithm that constructs an abstract nodule as a grayscale image, see the samples in Table 1. The function models nodules as elliptical shapes whose geometry, boundary, and intensity are determined by the six attributes. Spiculation is simulated via angular contour perturbations, edge sharpness by Gaussian blurring, and the optional internal structure by adding a small textured subregion. Random perturbations in rotation and background noise ensure slight natural variation across samples while preserving exact attribute control.

## 2.2. Customizability

The dataset framework is highly customizable, allowing systematic investigation of the impact of various factors on model performance. Target definitions can follow simple linear

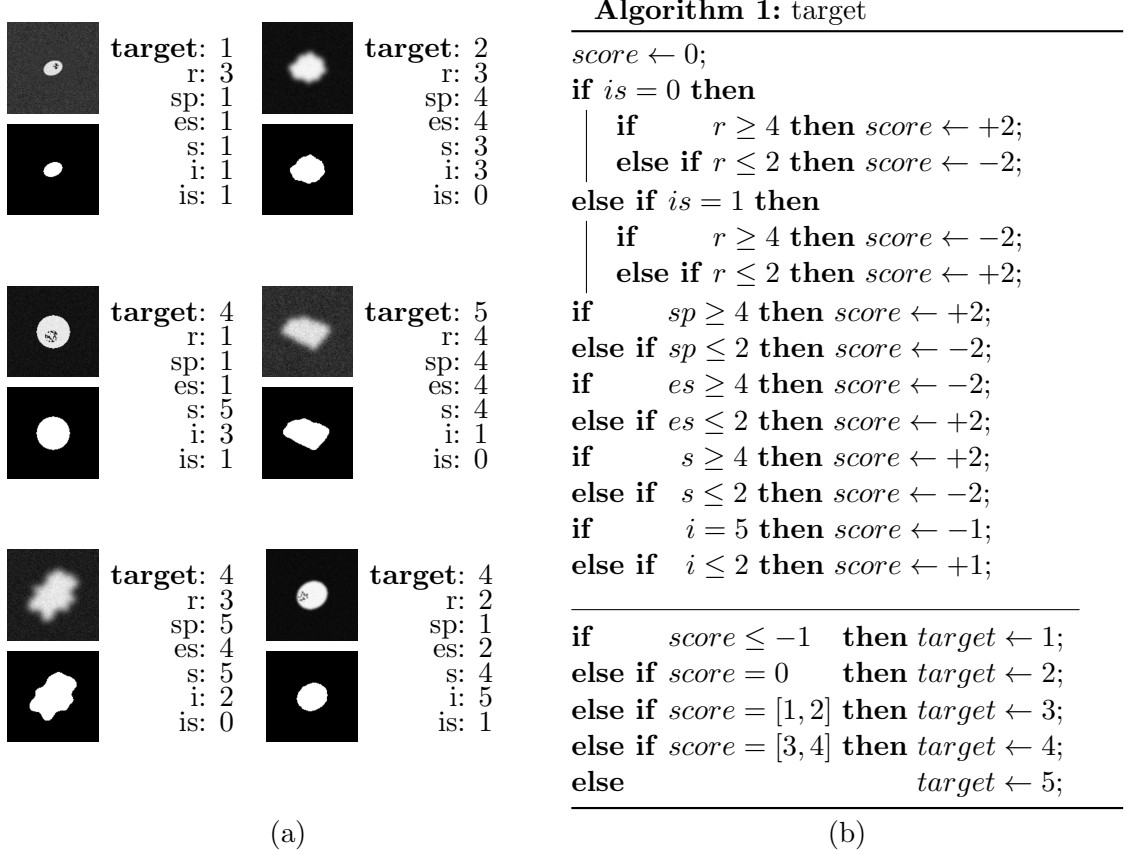

(a)

**Algorithm 1:** target

$score \leftarrow 0$;
**if** $is = 0$ **then**
    **if**      $r \geq 4$ **then** $score \leftarrow +2$;
    **else if** $r \leq 2$ **then** $score \leftarrow -2$;
**else if** $is = 1$ **then**
    **if**      $r \geq 4$ **then** $score \leftarrow -2$;
    **else if** $r \leq 2$ **then** $score \leftarrow +2$;
**if**      $sp \geq 4$ **then** $score \leftarrow +2$;
**else if** $sp \leq 2$ **then** $score \leftarrow -2$;
**if**      $es \geq 4$ **then** $score \leftarrow -2$;
**else if** $es \leq 2$ **then** $score \leftarrow +2$;
**if**      $s \geq 4$ **then** $score \leftarrow +2$;
**else if**   $s \leq 2$ **then** $score \leftarrow -2$;
**if**      $i = 5$ **then** $score \leftarrow -1$;
**else if**   $i \leq 2$ **then** $score \leftarrow +1$;

---

**if**      $score \leq -1$    **then** $target \leftarrow 1$;
**else if** $score = 0$     **then** $target \leftarrow 2$;
**else if** $score = [1, 2]$ **then** $target \leftarrow 3$;
**else if** $score = [3, 4]$ **then** $target \leftarrow 4$;
**else**                       $target \leftarrow 5$;

(b)

Table 1: **FunnyNodules** is a parametrized nodule-generation framework that enables full control and annotation of samples (a), as well as customizable image complexity. (b) Target rules are fully configurable and can represent complex attribute-correlated rules $(is, r)$ or simpler rules $(sp, es, s, i)$.

rules or be based on difficult attribute-correlated conditions, enabling analysis of model behavior under different levels of task complexity. The set of visual attributes can be varied in number, type, and scale. Experiments can be conducted with explicitly defined regions of interest (ROIs), and the presence of background introduces additional complexity in nodule detection and segmentation. Furthermore, image size can be adjusted to approximate realistic scaling, rather than relying on interpolation. Finally, input channels can be selected according to the simulated application domain, e.g., grayscale for chest X-ray or CT images and RGB for dermatological images, providing flexibility for diverse imaging tasks. While these examples illustrate key customization options, the framework is flexible and can accommodate additional variations.

## 3. Evaluation Methods

FunnyNodules offers broad opportunities for evaluating explainable AI methods, with the following examples illustrating the dataset's key capabilities. Building upon the definitions introduced by Nauta et al. (Nauta et al., 2023), we focus primarily on two complementary aspects of explanations, while acknowledging that others are also possible:

- Correctness: Measures how truthful an explanation is about the model's actual decision process, i.e., high correctness means the explanation reflects accurately what the model is doing.
- Contrastivity: Measures how well an explanation distinguishes the target outcome from other outcomes or events, i.e., high contrastivity means explanations highlight what makes the predicted class different from alternatives.

We used various models for our experiments, including standard models and prototype-based approaches. We trained ResNet-50 (He et al., 2016) and DenseNet-121 (Huang et al., 2017) in a multitask classification setting, where attributes and target predictions are concurrently predicted in the final layer. Models with hierarchical structures, such as Proto-Caps (Gallée et al., 2025b), HierViT (Gallée et al., 2025a), and Concept Bottleneck Model (joint setting) (Koh et al., 2020), naturally reflect the attribute-to-target relationship. We note that the models were not extensively optimized for the dataset, because the goal of this section is to demonstrate how FunnyNodules can be used for evaluation rather than to compare model performance.

### 3.1. Evaluation of a Model's Reasoning

The FunnyNodules dataset enables systematic assessment of whether a model correctly learns the relationships between diagnostic attributes and target classes through controlled variation of attribute values. By keeping the random seed and all other attributes constant, one can analyze how changes in a single attribute affect the target prediction, e.g., *If this nodule were more round, how would the model's prediction change?* This approach allows for the identification of incorrectly learned relationships or biases in the model's reasoning. As shown in Figure 1, 100 samples are generated for each attribute variation.

Figure 2 shows the mean target values for each attribute alongside the model predictions, which enables an assessment of how consistently the models follow the ground truth trends.

FunnyNodules also allows the investigation of more complex rules, such as the attribute roundness $(r)$. Its effect on the target depends on the presence of an internal structure $(is)$, as defined in Algorithm 1:

$$\text{target score} = \begin{cases} +2, & \text{if } (is = 0 \wedge r \geq 4) \text{ or } (is = 1 \wedge r \leq 2), \\ -2, & \text{if } (is = 0 \wedge r \leq 2) \text{ or } (is = 1 \wedge r \geq 4). \end{cases} \quad (1)$$

Using the parameterized generative algorithm of the FunnyNodules framework, we can systematically analyze attribute dependencies by isolating the effects of roundness and internal structure on the target, while fixing all other attribute values at 3 (see Figure 3).

Identifying attribute-dependent performance weaknesses is crucial for taking targeted measures, such as adapting model architectures or training procedures to increase sensitivity to specific attributes.

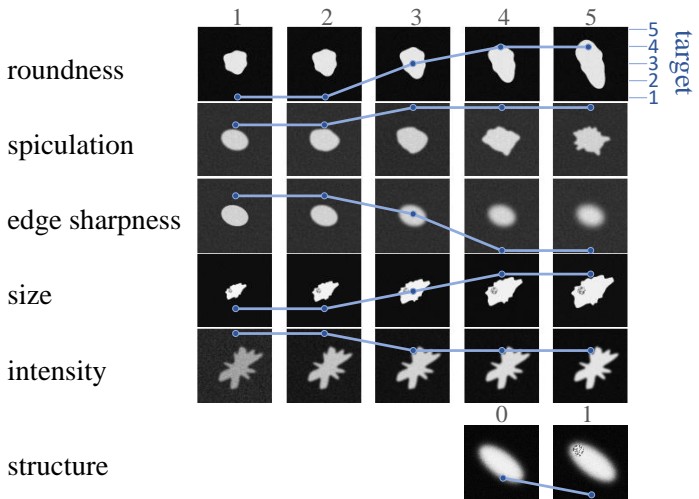

Figure 1: Controlled variation of a single attribute. Example nodules generated with varying values of one attribute are shown, together with an overlaid plot illustrating how these changes influence the target class (blue).

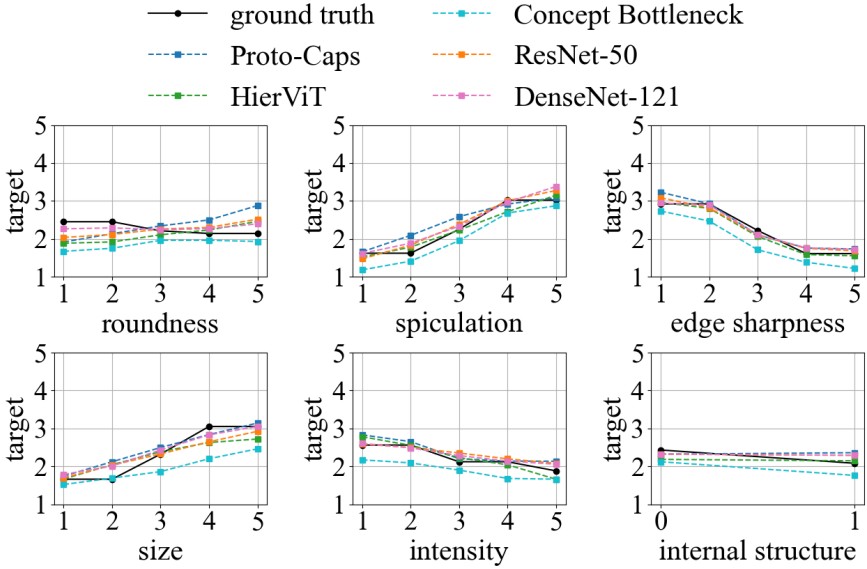

Figure 2: Sensitivity of model target predictions to variations in individual attributes. The figure illustrates how changes in predicted attribute values affect the predicted target class, providing insight into whether the underlying target rules are captured correctly. Most attributes are handled consistently across models, while the more complex notion of roundness shows notable deviations.

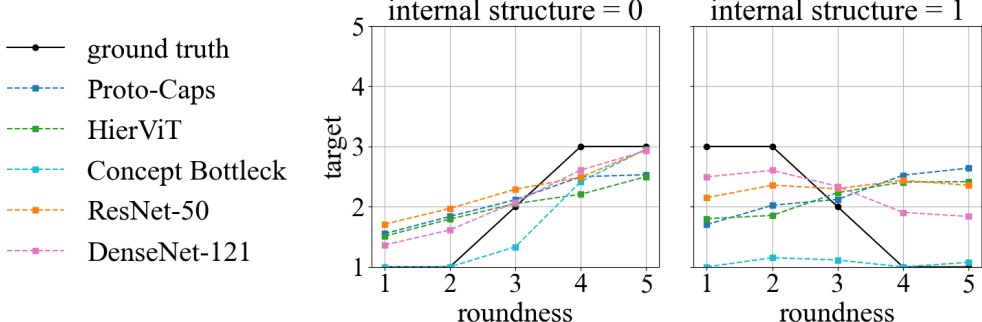

Figure 3: Evaluation of model behavior under a conditional decision rule with correlated attributes. The figure illustrates a scenario in which the effect of roundness on the target class depends on the presence of an internal structure. Results show that this conditional relationship is correctly captured only for the case of internal structure = 0, highlighting a general weakness of the tested models in handling correlated decision rules.

The controlled attribute-variation analyses explicitly capture the contrastivity of attribute explanations. By varying a single attribute while keeping all others fixed, we quantify how strongly that attribute shifts the model's predicted target, thereby revealing which factors drive the model to prefer one class over competing alternatives. In this setting, contrastivity is reflected in the magnitude of systematic changes in the predicted target distribution: large shifts indicate high contrastive relevance, whereas negligible shifts suggest low contrastivity.

Figure 2 and Figure 3 illustrate this effect by plotting target predictions as a function of a single attribute $a$. We define the contrastive prediction difference as

$$\Delta_{\text{target}}(a) = \mathbb{E}_x\big[f(x_{a=a_{\max}}) - f(x_{a=a_{\min}})\big], \tag{2}$$

where $f(\cdot)$ denotes the predicted target score and all non-varied attributes are held constant. This metric quantifies how strongly a model contrasts two otherwise identical inputs that differ only in one semantic attribute. For example, in analysis for Figure 2, when varying spiculation from its minimum (1) to maximum (5), the HierViT model exhibits a mean prediction difference of $\Delta_{\text{target}}(\text{spiculation})$=1.628 (ground truth: 1.4), indicating strong contrastive sensitivity to this attribute, which is consistent with the ground truth.

### 3.2. Investigating the Trustworthiness of Attribute-based Explanations

Attributes form the basis for explaining the model's target class prediction. Their correctness is an important measure for the truthfulness of the explanations and can be quantified in the same way as the target correctness. The relationship between these two performances provides insights into the model's reasoning process. To quantify it, we define a Trust Index

$$TI = P_{\text{target}} - \frac{\frac{1}{N}\sum_{i=1}^{N} A_i}{P_{\text{target}}}, \tag{3}$$

where $P_{target}$ is the target performance, $A_i$ denotes the performance for attribute $i$, and $N$ is the total number of attributes. The objective is to achieve a Trust Index (TI) close to zero. If $TI > 0$, the model exhibits strong target prediction performance, but the decisive attributes are not adequately learned. This suggests that the model's predictions are misguided and therefore should not be trusted. Conversely, if $TI < 0$, the model demonstrates strong attribute extraction capabilities, yet the mapping from attributes to the target class is insufficiently learned.

Table 2: **Trust Index (TI)** reflects both the reliability of a prediction and the correctness of its underlying decision rule. Positive TI values indicate low trustworthiness (e.g., orange), whereas negative TI values suggest an insufficiently learned target rule (e.g., pink).

| [train,val,test] | [100, 50, 500] | | [500, 50, 500] | | [1800, 200, 500] | |
| --- | --- | --- | --- | --- | --- | --- |
| | $P_{\text{target}}$ | $TI$ | $P_{\text{target}}$ | $TI$ | $P_{\text{target}}$ | $TI$ |
| ResNet-50 | 0.828 | -0.225 | 1.0 | 0.002 | 1.0 | 0.001 |
| DenseNet-121 | 0.824 | -0.242 | 1.0 | 0.083 | 0.998 | 0.009 |
| HierViT | 0.848 | -0.240 | 0.982 | -0.024 | 0.997 | -0.003 |
| Proto-Caps | 0.744 | -0.299 | 1.0 | 0.001 | 1.0 | 0.000 |
| Concept Bottleneck | 0.478 | -1.341 | 0.498 | -1.222 | 0.952 | -0.083 |

The Trust Index provides an immediate overview of the relationship between attribute and target prediction performance, see Table 2. It serves as a comparative diagnostic to reveal relative imbalances between attribute and target prediction, rather than as an absolute threshold-based measure. By varying the amount of training data, it can also be used to simulate different real-world data availability scenarios. For $TI > 0$, attribute extraction should be improved, for example by using differently weighted losses, whereas for $TI < 0$, the mapping from attributes to the target classes should be enhanced, for instance by employing more complex target layers.

### 3.3. Attribute ROI Masks for Attribute Attention Assessment

For models that perform attribute-based reasoning for diagnostic classification, evaluating attribute prediction becomes a key aspect. While standard evaluation metrics, such as the prediction scores presented above, provide an initial insight into model performance, a more in-depth analysis is required, analogous to the evaluation of target class predictions. One approach is to analyze the model's attention region that is most relevant for a prediction (Simonyan et al., 2014; Zeiler and Fergus, 2014). Applied at the attribute level, this allows the models to highlight the image regions that most strongly influenced its prediction for each attribute.

Assessing whether this attention is accurate requires ground truth annotations of the relevant regions of interest (ROIs). Creating real-world datasets with such attribute ROI annotations is highly challenging, particularly for medical images, as it requires manual labeling by experts. Although previous work (Choi et al., 2022) attempted automatic post-

hoc annotation of attribute-specific regions in real lung nodule datasets, it was limited to just two of the eight attributes.

In contrast, the procedurally generated FunnyNodules dataset enables fully controlled, parametric synthesis of attribute appearances and their corresponding annotations. This allows the generation of precise, attribute-specific ROIs directly during image creation, rather than relying on post-hoc segmentation, and thus provides exact and scalable ground truth for evaluating attribute-level attention, as illustrated in Figure 4.

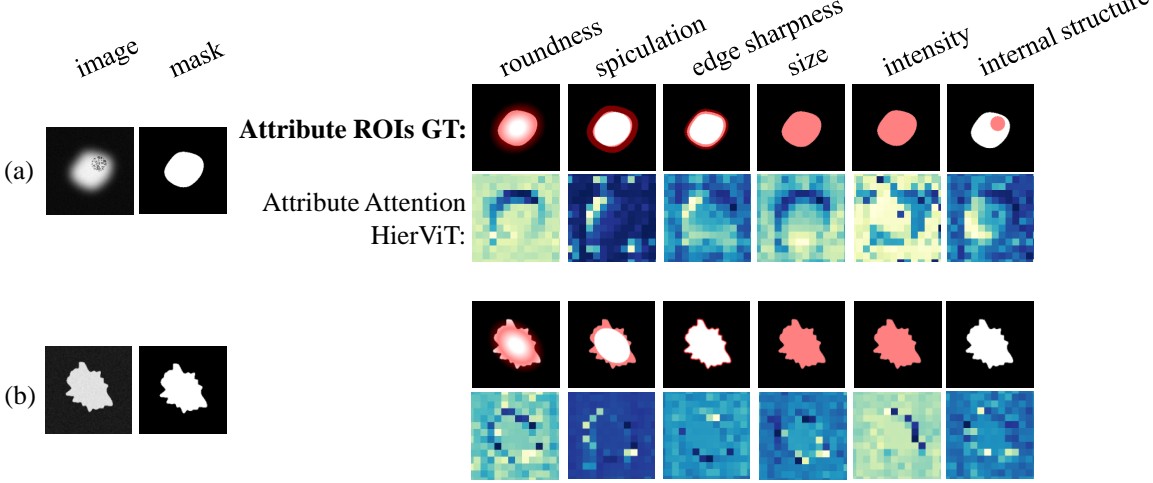

Figure 4: **Attribute ROIs.** Visualization of ground-truth attribute regions of interest (Attribute ROIs GT) generated during image synthesis. These regions can be used to evaluate spatial attention in attribute prediction by comparing them with model-derived attention maps (e.g., Attribute Attention HierViT), enabling assessment of alignment between predicted attention and ground-truth regions.

For roundness, the nodule's elliptical shape is the key structural feature. Spiculation is defined by the protruding spikes of the main nodule (samples b and c). Edge sharpness depends on the nodule border, with the attention ring narrowing for sharp edges (sample b, $es = 1$) and widening for soft edges (sample c, $es = 5$). Internal structure corresponds to the internal texture area (sample a). For attributes such as size and intensity, we hypothesize that the entire nodule is relevant for the model's prediction.

Figure 4 qualitatively compares HierViT attribute attention with ground-truth attribute ROIs. While HierViT attention heatmaps highlight the general nodule contours, comparison with ground-truth attribute ROIs reveals limited correspondence with the attribute-specific regions, indicating that attention is not tightly aligned with attribute localization.

### 3.4. Assessing Prototype Reasoning Correctness

Some xAI models leverage prototypes for explanation, as they resemble human reasoning (e.g., *This bird is an eagle because its beak resembles the eagle prototype*, or *this lung nodule*

*is malignant because its spiculation strongly resembles that of a learned prototype*). Compared to general-domain approaches that use object parts (Chen et al., 2019) or pixel values (Nauta et al., 2021) as explanatory features, this concept can be adapted to medical image diagnosis, with each prototype representing a single attribute as a decisive criterion (Gallée et al., 2025b,a). For an inference sample, the models present a prototype for each attribute, a training image closest to detected inference attribute value. The analysis of these prototype images provides insight into the performance of prototype learning and therefore the truthfulness of prototype reasoning. For more details about the prototype learning method, we refer to the respective works (Gallée et al., 2023, 2025a,b). We quantitatively assess the correctness of prototype-based reasoning summarized in Table 3. The results indicate how reliably the learned prototypes reflect ground-truth attributes and how faithfully target predictions can be reconstructed from prototype representations.

Table 3: Quantitative evaluation of prototype-based explanations. The table reports Within-1-Accuracy for attribute prototype correctness and prototype-induced target correctness. Attribute prototype correctness measures whether the selected prototype corresponds to the ground-truth attribute value, while prototype-induced target correctness denotes disease classification accuracy obtained when predictions are computed directly from the selected attribute prototypes in latent space. Results are reported on the test set of the data split [1800 train, 200 val, 500 test].

| | attribute prototypes | | | | | | attribute prototypes induced target |
| --- | --- | --- | --- | --- | --- | --- | --- |
| | r | sp | es | s | i | is | |
| HierViT | 0.974 | 0.998 | 0.880 | 0.960 | 1.0 | 0.964 | 0.977 |
| Proto-Caps | 0.982 | 0.915 | 0.945 | 0.863 | 0.995 | 0.979 | 0.988 |

Prototype-based explanations can likewise be used to evaluate contrastivity (Gurumoorthy et al., 2019; Van Looveren and Klaise, 2021). Beyond presenting the prototype nearest to an inference sample in latent space (i.e., the most similar example), contrastive prototypes corresponding to the largest latent distances from the inferred latent vector can be presented. Contrastive prototypes reveal the exemplar cases that the model considers maximally different from the input, thereby highlighting which attribute appearance would tip the model toward an alternative diagnosis.

### 3.5. Scalable Dataset Size for Unlimited Evaluation

Dataset size is a fundamental limitation in medical imaging. The sensitive nature of patient data restricts the total number of images available, and the specialized expertise required for annotation further constrains the size of fully annotated datasets. While disease labels can be extracted automatically from clinical reports (Smit et al., 2020), key attributes for reasoning alignment in AI models must be annotated manually (Gallée et al., 2026). This process is costly and further limits the availability of real-world datasets.

Consequently, evaluating model robustness under dataset constraints using real data is inherently limited. In contrast, synthetic datasets, such as FunnyNodules, provide virtually unlimited scalability. As demonstrated in Table 2, the Concept Bottleneck model is highly sensitive to sample size, impacting both predictive performance and the quality of explanations. Such findings pose key considerations when selecting models.

### 3.6. Further Evaluation Ideas

Beyond the presented approaches for using the dataset in model evaluation, more evaluation strategies can be considered, such as:

*Background independence* In real medical images, background structures can interfere with nodule detection. The FunnyNodules framework allows the controlled addition of background structures, enabling systematic evaluation of models' robustness to such interference.

*Model-specific evaluation* Because image generation is fully controlled, differences in model activations can be systematically observed and analyzed, enabling identification of the internal representations affected by specific attribute variations.

## 4. Discussion and Conclusion

In this work, we introduce the FunnyNodules dataset, which provides extensive opportunities for evaluating diverse aspects of model behavior and explainability methods, with a particular focus on attribute-based reasoning models. The fully controlled parametric synthesis of attribute characteristics enables the creation of multiple types of ground truth annotations, such as attribute-level scores and region-of-interest masks. Moreover, the framework's high customizability allows precise specification of how attributes relate to target classes, making it possible to simulate different levels of reasoning complexity and decision rules. We also provided example analyses highlighting the dataset's versatility for evaluating and interpreting AI models. As a scalable benchmark, the dataset supports future comprehensive, method-specific evaluations.

The controlled conditions of FunnyNodules offer clear advantages in terms of scalability, reproducibility, and precise ground truth. While the dataset is neither designed nor intended to replace evaluation on real-world data, which remains a clear limitation, it enables forms of systematic analysis that are difficult or impossible with real medical data due to inherent uncertainty, heterogeneity, and incomplete annotation. In particular, FunnyNodules is not intended to model medically complete or semantically valid lung nodule attributes, and evaluation results obtained on this synthetic dataset are therefore not directly transferable to real data at the semantic level. Nevertheless, the dataset allows controlled variation of factors such as visual attribute complexity, target rules, data availability, and annotation richness, enabling systematic investigation of model behavior under well-defined conditions. This makes it possible to study how different model architectures and learning strategies respond to real-world challenges. While absolute performance may not translate to real-world applications, the resulting insights into internal model mechanisms are informative for model development and comparison.

Comprehensive evaluation of explanations generated by xAI models is essential for their intended use in practice. A central component of such evaluation is the inclusion of humans

in the loop (Dieber and Kirrane, 2022; Rong et al., 2023; Gallée et al., 2024), as explainability ultimately targets human-centered aspects such as understanding, acceptance, and usability. However, conducting user studies—particularly in medical applications—requires access to domain experts and involves substantial effort and cost. In this context, datasets such as FunnyNodules provide a complementary evaluation tool by enabling scalable and objective analyses of explanation correctness and faithfulness under fully controlled conditions. While such dataset-driven evaluations cannot replace human studies, which remain essential for assessing clinical relevance and real-world utility, they can substantially reduce the evaluation burden and support systematic analysis of model behavior. The FunnyNodules framework thus offers a versatile foundation for advancing the evaluation of explainable AI methods and contributes to the development of more transparent and trustworthy medical AI systems.

## Acknowledgments

This study was supported by the German Federal Ministry of Research, Technology and Space BMFTR as part of the University Medicine Network (Project: RACOON, 01KX2121) and by the German Research Foundation DFG (Project: KEMAI, GRK 3012 – 520750254).

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

## Appendix A. FunnyNodules

### A.1. Histogram

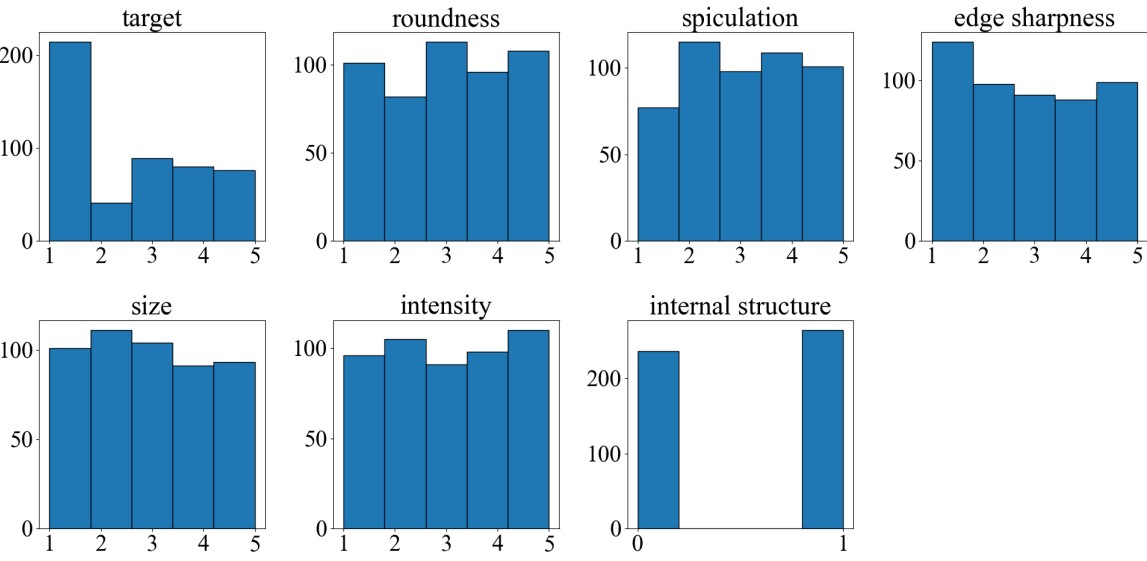

Figure 5: Histogram of 500 randomly generated FunnyNodules images.

