# OpenReview forum: "FunnyNodules: A Customizable Medical Dataset Tailored for Evaluating Explainable AI"
_MIDL.io/2026/Conference — MIDL 2026 Poster_

### Official Review · Reviewer_HAmK · 2026-01-09

**Confidence:** 5
**Preliminary Rating:** 4
**Final Rating:** 5

**Summary:**

The authors introduce FunnyNodules, a synthetic dataset designed for interrogating concept-based XAI methods applied to the medical domain. The images are based on lung nodules and the relationship between the concepts present and the class can be chosen in a complex manner similar to lung nodules in CT imaging. The authors test multiple black-box and interpretable models on their dataset and give example comparisons for the evaluation of different aspects of interpretability.

**Strengths:**

The dataset is novel, well designed, and well suited for the evaluation of different aspects of interpretability. The evaluation gives readers some example ways that methods could be compared. The additional dense labels (masks) provided for each concept could lead to some interesting analysis when comparing methods, e.g., saliency maps, or the similarity scores of prototype-based models.

**Weaknesses:**

The actual insights/evaluations provided in this paper are uninspiring. Very few conclusions are made comparing different methods. An exception here is the statement on the Concept Bottleneck model being sensitive to training data size on page 9. More statements/comparisons like this would be desirable. The lack of a more comprehensive comparison is understandable given the aim of the paper is to introduce the dataset, but the paper would be strengthened with an in-depth discussion/comparison of multiple XAI methods. Likewise, the authors introduce the concept masks but perform no analyses with them.

**Detailed Comments:**

A TMLR paper introduced Elements, a synthetic dataset specifically designed for understanding concept-based interpretability methods. This would be a good reference to include and discuss in relation to FunnyNodules. https://arxiv.org/abs/2404.03713

"Histogram analysis of FunnyNodules can be found in Appendix A.1" This makes sense once a reader checks the appendix, but the sentence in isolation does not.

"TI << 0 suggests an insufficiently learned target rule" -0.083 is highlighted as much less than 0, but -0.08 does not seem like much less than 0. A discussion on what is considered a substantial difference between methods would be appreciated as the meaning of the magnitude of the metric is not easily intuited.

Table 3 contains Prototype Correctness, which is not clearly defined mathematically. As a reader, I can guess what it means, but this should be explicitly stated.

"Datasets such as FunnyNodules provide a complementary approach that reduces this burden, as no human study is needed" It is too strong a statement to say human studies are not needed. The authors are correct to state that synthetic datasets provide a valuable tool in the evaluation pipeline of an XAI method, but they cannot and should not be used to entirely replace human studies. Some aspects of interpretability simply require a human to test.

The link between the stated focus on contrastivity (page 5) and the results presented in the paper is unclear.

**Justification Of Final Rating:**

The authors have adequately addressed all comments made in my review and I have adjusted my final score accordingly. I still think there are areas where the paper could be improved (e.g. The quantitative value of the proposed Trust Index is hard to understand, or, how can we concretely define a ground truth saliency map for the concept of size? The whole object, or the edges? Or is it just not possible?), but all papers can be improved somehow.

**Justification Of The Preliminary Rating:**

The concept of the paper is good and the dataset interesting, but some of the analysis is unclear and a more comprehensive analysis/discussion comparing different XAI methods could have been performed.

**Questions To Address In The Rebuttal:**

Please address the comments in "Detailed Comments".

---

> ### Author Response · Authors · 2026-01-23
>
> We thank the Reviewer for the positive comments and the constructive feedback.
>
> We appreciate the interpretation of this paper as an introduction to the synthetic dataset and agree that more extensive experiments would provide additional insights into model behavior. However, we intentionally avoid strong performance rankings between methods, because this dataset paper cannot provide the comprehensive and scientifically rigorous experimental optimization and documentation required for definitive comparisons, which could otherwise lead to misleading or even false findings. To avoid conveying misleading impressions, we explicitly state that the models are not extensively optimized and that the presented experiments serve as illustrative examples of the types of systematic analyses enabled by the dataset. In the revised manuscript, we further clarify this scope and emphasize that comprehensive method comparisons are an important direction for future work. Additionally, we added an exemplary analysis of concept masks (see next point).
>
> We address the detailed comments point by point:
>
> **1. Concept masks are not analyzed**
> In Figure 4, we added illustrative attention heatmaps (e.g., from HierViT) to demonstrate how attribute ROIs can be used to assess the alignment between model attention and ground-truth regions.
>
> **2. Reference to the “Elements” dataset**
> Thank you for pointing out this work. We added ‘Elements’ as a related synthetic dataset in the Introduction and clarified how FunnyNodules complements prior work by focusing on lesion-specific attributes in medical imaging rather than natural images or abstract primitives.
>
> **3. “Histogram analysis …” sentence is isolated**
> We revised the sentence to clarify that the histogram analysis illustrates the distribution of attribute and target labels in FunnyNodules.
>
> **4. TI ≪ 0, TI ≫ 0**
> We agree that this notation overstates the interpretation given the observed magnitudes. In the revised manuscript, we replaced it with TI < 0 and TI > 0 and clarified that the Trust Index is intended as a comparative diagnostic rather than an absolute measure.
>
> **5. Table 3 prototype correctness not clearly defined**
> We updated the caption to clearly define how prototype correctness values are computed, including the metric used.
>
> **6. Overstatement about human studies**
> We fully agree that human studies are essential for the evaluation of xAI methods, and we emphasize this in the corresponding paragraph. Certain aspects of explainability evaluation can be addressed through dataset-driven analyses, for which synthetic datasets provide a complementary tool offering scalability and controllability, and enabling detailed inspection of model behavior under well-defined conditions. Other aspects, such as usability, acceptance, and understanding of model outputs and reasoning, can only be evaluated with humans in the loop. We have revised the manuscript accordingly to remove the misleading statement and to clearly position FunnyNodules as a complementary evaluation tool rather than a replacement for user studies.
>
> **7. Contrastivity not represented in results**
> We agree that contrastivity was underrepresented in the initial submission. In the revised version, we describe in Section 3.1 (controlled-variation analyses) and Section 3.4 (prototype reasoning) how contrastivity can be assessed.

---

### Official Review · Reviewer_1Uid · 2026-01-10

**Confidence:** 4
**Preliminary Rating:** 3
**Final Rating:** 4

**Summary:**

This paper proposes FunnyNodules, a synthetic dataset designed for systematic analysis of attribute-based reasoning in medical AI models. Images are generated according to 6 attributes (roundness, spiculation, edge sharpness, size, intensity, internal structure), corresponding to lung nodule–like shapes and attributes, and various neural networks are used to predict the generation parameters. ROI masks are generated for attribute attention assessment. Explainable AI evaluation focuses primarily on two aspects (correctness, contrastivity). A trust index is given to evaluate the relationship between these two performances provides insights into the model’s reasoning process. Various models demonstrate the ability to predict the various attributes to varying degrees.

**Strengths:**

Its advantageous to generate synthetic data, for data-hungry training processes.
Figure 2 seems to indicate the various networks were able to predicting the generation parameters to various degrees, ViT is good for roundness.
********************************************************************************************************************************

**Weaknesses:**

It’s not clear how useful this dataset or synthetic data methodology will be in real data scenarios. Could it be used to evaluate and interpret lung nodule attributes in real data? How easily could the methodology adapt to interpreting classification in other data contexts?

**Detailed Comments:**

See above comments

**Justification Of Final Rating:**

Rebuttal answered my questions.
************************************************************************************************************************************************************************************

**Justification Of The Preliminary Rating:**

An effort is made to simulate data mixing 6 different properties related to lung nodules, however it is not clear how practically or generally useful the method is on real data.
**************************

**Questions To Address In The Rebuttal:**

See above comments regarding practical applicability to real data and contexts other than lung-nodule like scenarios.

---

> ### Author Response · Authors · 2026-01-23
>
> We thank the Reviewer for the positive comments on the dataset’s comprehensive annotation and would like to address the concerns regarding the practicability and usefulness of the dataset for evaluating models intended for real-world data.
>
> The primary goal of this work is to improve the understanding of (explainable) AI approaches by presenting a synthetic dataset with fully known and controllable ground truth. The dataset can be adapted to specific model evaluation tasks, but is not designed to resemble real lung nodules, even though they were used as a source of inspiration. We intentionally do not aim to model medically correct target definitions, completeness of relevant attributes, or realistic visual appearance. Consequently, performance evaluation results obtained on this synthetic dataset are not directly transferable to real data at the semantic level.
>
> The dataset allows controlled variation and systematic investigation of model behavior with respect to factors such as the complexity of visual attributes and the target rules, data availability, and annotation richness. For example, the conditional decision rules shown in Figure 3 are common in radiology [1], where diagnostic criteria interact with each other. Different model architectures and learning strategies can be tested extensively on synthetic data to assess which settings can learn such complex, context-dependent effect on the target. Such datasets can be simulated to conduct feasibility tests for model training.
>
> We believe that the proposed dataset framework for model analysis can be transferred to settings beyond lung nodule classification. Importantly, this approach is designed to analyze model behavior rather than the information content of the data itself. Insights into model behavior, such as sensitivity to training data size (Table 2) or to the complexity of attribute–target relationships (Section 3.1), are therefore transferable to models trained on real-world data. While such analyses can and should also be conducted using real datasets, evaluating attribute-based reasoning processes requires extensive ground-truth annotation of the relevant diagnostic criteria. With FunnyNodules, we aim to help bridge this gap and support further research on attribute-driven AI models.
>
> [1] MacMahon, Heber, et al. "Guidelines for management of incidental pulmonary nodules detected on CT images: from the Fleischner Society 2017." Radiology (2017): 228-243.

---

### Official Review · Reviewer_F2C5 · 2026-01-15

**Confidence:** 3
**Preliminary Rating:** 3
**Final Rating:** 4

**Summary:**

The authors propose FunnyNodules, a customizable synthetic vision dataset for evaluating explainable AI by generating nodule-like images with controllable attributes. To demonstrate how the dataset can assess model behavior, the authors run controlled experiments that vary one attribute at a time and test conditional decision rules. They validate these evaluations using quantitative checks such as Trust Index, ROI-based alignment analyses for attention, and prototype-based correctness measures, showing that different models behave differently under the same ground-truth rules.

**Strengths:**

This paper addresses the gap of the lack of comprehensive ground truth information in medical imaging, enabling more systematic evaluation of xAI methods than is typically possible with real clinical data. The dataset provides not only target labels but also attribute labels and attribute-specific ROI masks, which are rare in medical imaging benchmarks.

**Weaknesses:**

- The author mentioned two “complementary aspects of explanations” at the beginning of section 3: correctness and contrastivity. While the correctness is well established and quantified in the later section, further statements on contrastivity are missing. Beyond the brief definition, contrastivity is not further developed, and it is not explicitly discussed in the rest of the paper.

- In section 3.4, table 3 is placed here, but there is no reference to it. Table 3’s caption is largely declarative rather than descriptive, and the table lacks a clear definition of what each column’s numbers represent. Adding a precise metric and interpretation in the caption or surrounding text would substantially improve clarity.

**Detailed Comments:**

- The same caption issue described before applies to some other figures and tables. A caption should be self-contained enough to interpret the table/figure quickly, by briefly defining reported metrics, clarifying what the axes/columns represent, and avoiding vague phrasing.

**Justification Of Final Rating:**

The authors have addressed all of my questions and concerns in the rebuttal, and their clarifications resolve the issues I previously raised. As a result, I am raising my rating to Weak Accept. Overall, the revised manuscript reads as sufficiently strong and well-justified, and I do not have any major concerns.

**Justification Of The Preliminary Rating:**

The dataset idea is promising and provides rare comprehensive ground truth information that is useful for evaluating xAI in a way that real medical datasets often can’t. However, the major concern is that the mentioned "contrastivity" is not well explained and demonstrated in the paper, and some captions for the tables/figures are hard to interpret because they are not descriptive enough.

**Questions To Address In The Rebuttal:**

- Please consider addressing the points in “weaknesses” and “detailed comments”.

---

> ### Author Response · Authors · 2026-01-23
>
> We thank the Reviewer for the positive assessment of our main contribution and for identifying aspects of the manuscript that required improvement.
> We appreciate the acknowledgment of the current lack of comprehensive annotated datasets for evaluating xAI methods, as well as the recognition that the extensive annotation of our synthetic dataset provides a foundation for more systematic evaluation.
> Below, we address the Reviewer’s comments point by point.
>
> **1. Contrastivity**
> We agree that in the initial submission, contrastivity was introduced at a conceptual level but not sufficiently operationalized. In the revised manuscript, we explicitly clarify how contrastivity is captured within our framework:
>
> • In Section 3.1, we now describe how controlled attribute variation analyses, depicted in Figures 2 and 3, operationalize contrastivity by examining how systematic perturbations of individual attributes affect the model's predictions. We explicitly quantify contrastivity by presenting the expected change in the predicted target metric and illustrating it with an example.
>
> • In Section 3.4, we extend the discussion of prototype-based explanations to explicitly include contrastive prototypes. In addition to nearest (supporting) prototypes, we now describe how identifying the most dissimilar prototypes in latent space highlights exemplar cases that the model considers maximally different from the input.
>
> **2. Table 3 reference and caption clarity**
> We revised Section 3.4 to explicitly reference Table 3 in the main text and to interpret its results in the context of prototype reasoning correctness. Furthermore, we reworked the caption of Table 3, as well as those of other figures and tables, to ensure they are fully self-contained and clearly describe the information presented.

---

### Author Rebuttal · Authors · 2026-01-23

**Rebuttal:**

We thank all Reviewers for their detailed and constructive comments, which helped us improve the manuscript. We are pleased that the Reviewers agree that FunnyNodules is well suited for evaluating diverse aspects of model interpretability. As noted, the lack of dense, attribute-level annotations in real medical datasets limits the systematic evaluation of xAI methods. By providing fully controlled images with rich ground-truth information, FunnyNodules offers a complementary tool for probing model reasoning and explanation behavior.

The Reviewers’ critique points were fair, and we addressed underrepresented aspects of contrastivity as well as revised the captions of tables and figures to improve clarity (F2C5, HAmK). We also addressed concerns regarding the practicability and usefulness of the dataset for evaluating models on real data (1Uid) and further clarified the cautious framing of model performance comparisons in the experiments (HAmK). We address the detailed comments point by point for each reviewer.

We believe these revisions strengthen the manuscript and further highlight the dataset’s potential to support systematic evaluation of explainable AI in medical imaging.

**Supporting Material:**

/attachment/16f7d9248f2f3811769ef5c8a2a7e4bdea73b849.zip

---

### Comment · Area_Chair_StGV · 2026-01-29
**Discussion phase**

Dear Reviewers, the manuscript has now entered the discussion phase.

We kindly invite you to evaluate the authors’ responses and the revised manuscript, and to engage in discussion with the authors to address any remaining questions or unresolved points.

Once you have completed your review and discussion, submit your final rating. Please complete this step by selecting “Edit” → “Official Review” no later than February 1, 2026, at 23:59 AoE.

---

### Meta-Review · Area_Chair_StGV · 2026-02-07

**Recommendation:** Accept (Oral)
**Confidence:** 5

**Metareview:**

This paper is a clear acceptance. The reviewers unanimously found the proposed dataset to be valuable for the systematic evaluation of interpretability methods. Additionally, the rebuttal was well-received, leading to increased ratings across the board.

---

### Decision · Program_Chairs · 2026-02-13

Accept (Poster)